# INFOGRAPH: UNSUPERVISED AND SEMI-SUPERVISED GRAPH-LEVEL REPRESENTATION LEARNING VIA MUTUAL INFORMATION MAXIMIZATION

**Fan-Yun Sun[1,2], Jordan Hoffmann[2,4], Vikas Verma [2,3], Jian Tang[2,5,6]**
[1]National Taiwan University,
[2]Mila-Quebec Institute for Learning Algorithms, Canada
[3]Aalto University, Finland
[4]Harvard University, USA
[5]HEC Montreal, Canada
[6]CIFAR AI Research Chair

```
b04902045@ntu.edu.tw
jhoffmann@g.harvard.edu
vikas.verma@aalto.fi
jian.tang@hec.ca
```

## ABSTRACT

This paper studies learning the representations of whole graphs in both unsupervised and semi-supervised scenarios. Graph-level representations are critical in a variety of real-world applications such as predicting the properties of molecules and community analysis in social networks. Traditional graph kernel based methods are simple, yet effective for obtaining fixed-length representations for graphs but they suffer from poor generalization due to hand-crafted designs. There are also some recent methods based on language models (e.g. graph2vec) but they tend to only consider certain substructures (e.g. subtrees) as graph representatives. Inspired by recent progress of unsupervised representation learning, in this paper we proposed a novel method called InfoGraph for learning graph-level representations. We maximize the mutual information between the graph-level representation and the representations of substructures of different scales (e.g., nodes, edges, triangles). By doing so, the graph-level representations encode aspects of the data that are shared across different scales of substructures. Furthermore, we further propose InfoGraph*, an extension of InfoGraph for semi-supervised scenarios. InfoGraph* maximizes the mutual information between unsupervised graph representations learned by InfoGraph and the representations learned by existing supervised methods. As a result, the supervised encoder learns from unlabeled data while preserving the latent semantic space favored by the current supervised task. Experimental results on the tasks of graph classification and molecular property prediction show that InfoGraph is superior to state-of-the-art baselines and InfoGraph* can achieve performance competitive with state-of-the-art semi-supervised models.

## 1 INTRODUCTION

Graphs have proven to be an effective way to represent very diverse types of data including social networks Newman & Girvan (2004), biological reaction networksPavlopoulos et al. (2011), protein-protein interactions Krogan et al. (2006), the quantum mechanical properties of individual molecules Xie & Grossman (2018); Jin et al. (2018), and many more. Graphs provide explicit information about the coupling between individual units in a larger part along with a well defined framework for assigning properties to the nodes and the edges connecting them. There has been a significant amount of previous work done studying many aspects of graphs including link prediction Gao et al. (2011); Wang et al. (2011) and node prediction Blei et al. (2003). Due to its flexibility, graph-like data structures can capture rich information which is critical in many applications.

At the lowest level, much work has been done on learning node representations– low-dimensional vector embeddings of individual nodes Perozzi et al. (2014); Tang et al. (2015); Grover & Leskovec (2016). Another field that has attracted a large amount of attention recently is learning representations of entire graphs. Such a problem is critical in a variety of applications such as predicting the properties of molecular graphs in both drug discovery and material science Chen et al. (2019b;a). There has been some recent progress based on neural message passing algorithms Gilmer et al. (2017); Xie & Grossman (2018), which learn the representations of entire graphs in a supervised way. These methods have been shown achieving state-of-the-art results on a variety of different prediction tasks Kipf et al. (2018); Xie & Grossman (2018); Gilmer et al. (2017); Chen et al. (2019a).

However, one of the most difficult obstacles for supervised learning on graphs is that it is often very costly or even impossible to collect annotated labels. For example, in the chemical domain labels are typically produced with a costly Density Functional Theory (DFT) calculation. One option is to use semi-supervised methods which combine a small handful of labels with a larger, unlabeled, dataset. In real-world applications, partially labeled datasets are common, making tools that are able to efficiently utilize the present labels particularly useful.

Coming up with methods that are able to learn unsupervised representations of an entire graph, as opposed to nodes, is an important step in working with unlabeled or partially labeled graphs Narayanan et al. (2017); Hu et al. (2019); Nguyen et al. (2017). For example, there exists work that explores pre-training techniques for graphs to improve generalization Hu et al. (2019). Another common approach to unsupervised representation learning on graphs is through graph kernels Pržulj (2007); Kashima et al. (2003); Orsini et al. (2015). However, many of these methods do not provide explicit graph embeddings which many machine learning algorithms operate on. Furthermore, the handcrafted features of graph kernels lead to high dimensional, sparse or non-smooth representations and thus result in poor generalization performance, especially on large datasets Narayanan et al. (2017).

Unsupervised learning of latent representations is also an important problem in other domains, such as image generation Kingma & Welling (2013); Kim & Mnih (2018) and natural language processing Mikolov et al. (2013a). A recent work introduced Deep Infomax, a method that maximizes the mutual information content between the input data and the learned representation Hjelm et al. (2018). This method outperforms other methods on many unsupervised learning tasks. Motivated by Deep InfoMax Hjelm et al. (2018), we aim to use mutual information maximization for unsupervised representation learning on the entire graph. Specifically, our objective is to maximize the mutual information between the representations of entire graphs and the representations of substructures of different granularity. We name our model **InfoGraph**.

We also propose a semi-supervised learning model which we name **InfoGraph\***. We employ a student-teacher framework similar to Mean-Teacher method Tarvainen & Valpola (2017). We maximize the mutual information between intermediate representations of the two models so that the student model learns from the teacher model. The student model is trained on the labeled data using a supervised objective function while the teacher model is trained on unlabeled data with **InfoGraph**. Using **InfoGraph\***, we achieve performance competitive with state-of-the-art methods on molecular property prediction.

We summarize our contributions as follows:

- We propose InfoGraph, an unsupervised graph representation learning method based on Deep InfoMax (DIM) Hjelm et al. (2018).

- We show that InfoGraph can be extended to semi-supervised prediction tasks on graphs.

- We empirically show that InfoGraph surpasses state-of-the-art performance on graph classification tasks with unsupervised learning and obtains performance comparable with state-of-art methods on molecular property prediction tasks using semi-supervised learning.

## 2 RELATED WORK

Representation learning for graphs has mainly dealt with supervised learning tasks. Recently, however, researchers have proposed algorithms that learn graph-level representations in an unsupervised manner Narayanan et al. (2017); Adhikari et al. (2018).

Concurrently to this work, information maximizing graph neural networks (IGNN) was introduced which uses mutual information maximization between edge states and transform parameters to achieve state-of-the-art predictions on a variety of supervised molecule property prediction tasks Chen et al. (2019b). In this work, our focus is on unsupervised and semi-supervised scenarios.

**Graph Kernels**. Constructing graph kernels is a common unsupervised task in learning graph representations. These kernels are typically evaluated on node classification tasks. In graph kernels, a graph $G$ is decomposed into (possibly different) $\{G_s\}$ sub-structures. The graph kernel $K(G_1, G_2)$ is defined based on the frequency of each sub-structure appearing in $G_1$ and $G_2$ respectively. Namely, $K(G_1, G_2) = \langle f_{G_{s_1}}, f_{G_{s_2}} \rangle$, where $f_{G_s}$ is the vector containing frequencies of $\{G_s\}$ sub-structures, and $\langle, \rangle$ is an inner product in an appropriately normalized vector space. Much work has been devoted to deciding which sub-structures are more suitable than others (refer to appendix A.1). Instead of defining hand crafted similarity measures between substructures, InfoGraph adopts a more principled metric – mutual information.

**Contrastive methods**. An important approach for unsupervised representation learning is to train an encoder to be contrastive between representations that capture statistical dependencies of interest and those that do not. For example, a contrastive approach may employ a scoring function, training the encoder to increase the score on "real" input (a.k.a, positive examples) and decrease the score on "fake" input (a.k.a., negative samples). For more detailed discussion, refer to appendix A.2.

*Deep Graph InfoMax* (DGI) Veličković et al. (2018) also belongs to this category, which aims to train a node encoder that maximizes mutual information between node representations and the pooled global graph representation. Although we built upon a similar methodology, our aim is different than theirs as our goal is to obtain embeddings at the whole graph level for unsupervised and semi-supervised learning whereas DGI only evaluates node level embeddings. In order to differentiate our method with Deep Graph Infomax (Veličković et al. (2018)), we term our model InfoGraph.

**Semi-supervised Learning**. A comprehensive overview of semi-supervised learning (SSL) methods is out of the scope of this paper. We refer readers to Appendix B for a short overview or Zhu et al. (2003); Chapelle et al. (2006); Oliver et al. (2018) for more comprehensive discussions. Here, we discuss a state-of-the-art method applicable for regression tasks – Mean Teacher Tarvainen & Valpola (2017).Mean Teacher adds a loss term which encourages the distance between the original network's output and the teacher's output to be small. The teacher's predictions are made using an exponential moving average of parameters from previous training steps. Inspired by the "student-teacher" framework in Mean Teacher model, our semi-supervised model (InfoGraph*) deploys two separate encoders but instead of explicitly encouraging the output of the student model to be similar to the teacher model's output, we enable the student model to learn from the teacher model by maximizing mutual information between intermediate representations learned by two models.

## 3 METHODOLOGY

Most recent work on graphs focus on supervised learning tasks or learning node representations. However, many graph analytic tasks such as graph classification, regression, and clustering require representing entire graphs as fixed-length feature vectors. Though graph-level representations can be obtained through the node-level representations implicitly, explicitly extracting the graph can be more straightforward and optimal for graph-oriented tasks.

Another scenario that is important, yet attracts comparatively less attention in the graph related literature is semi-supervised learning. One of the biggest challenges in prediction tasks in biology Yan et al. (2017); Yang et al. (2014) or molecular machine learning Duvenaud et al. (2015); Gilmer et al. (2017); Jia & Liang (2017) is the extreme scarcity of labeled data. Therefore, semi-supervised learning, in which a large number of unlabeled samples are incorporated with a small number of labeled samples to enhance accuracy of models, will play a key role in these areas.

In this section, we first formulate an unsupervised whole graph representation learning problem and a semi-supervised prediction task on graphs. Then, we present our method to learn graph-level representations. Afterwards we present our proposed model for the semi-supervised learning scenario.

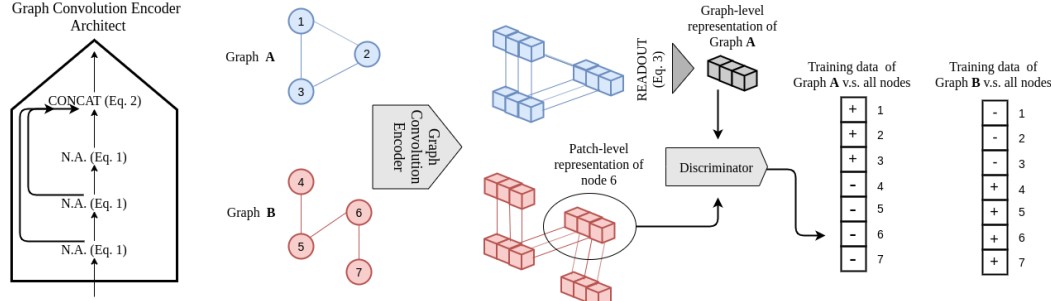

Figure 1: Illustration of InfoGraph. N.A. denotes neighborhood aggregation. An input graph is encoded into a feature map by graph convolutions and jumping concatenation. The discriminator takes a (global representation, patch representation) pair as input and decides whether they are from the same graph. InfoGraph uses a batch-wise fashion to generate all possible positive and negative samples. For example, consider the toy example with 2 input graphs in the batch and 7 nodes (or patch representations) in total. For the global representation of the blue graph, there will be 7 input pairs to the discriminator and same for the red graph. Thus, the discriminator will take 14 (global representation, patch representation) pairs as input in this case.

### 3.1 PROBLEM DEFINITION

**Unsupervised Graph Representation Learning**. Given a set of graphs $\mathbb{G} = \{G_1, G_2, ...\}$ and a positive integer $\delta$ (the expected embedding size), our goal is to learn a $\delta$-dimensional distributed representation of every graph $G_i \in \mathbb{G}$. We denote the number of nodes in $G_i$ as $|G_i|$. We denote the matrix of representations of all graphs as $\Phi \in \mathbb{R}^{|\mathbb{G}| \times \delta}$.

**Semi-supervised Graph Prediction Tasks**. Given a set of labeled graphs $\mathbb{G}^L = \{G_1, \cdots, G_{|\mathbb{G}^L|}\}$ with corresponding output $\{o_1, \cdots, o_{|\mathbb{G}^L|}\}$, and a set of unlabeled samples $\mathbb{G}^U = \{G_{|\mathbb{G}^L|+1}, \cdots, G_{|\mathbb{G}^L|+|\mathbb{G}^U|}\}$, our goal is to learn a model that can make predictions for unseen graphs. Note that in most cases $|\mathbb{G}^U| \gg |\mathbb{G}^L|$.

### 3.2 INFOGRAPH

We focus on graph neural networks (GNNs)—a flexible class of embedding architectures which generate node representations by repeated aggregation over local node neighborhoods. The representations of nodes are learned by aggregating the features of their neighborhood nodes, so we refer to these as patch representations. GNNs utilize a READOUT function to summarize all the obtained patch representations into a fixed length graph-level representation.

Formally, the $k$-th layer of a GNN is

$$h_v^{(k)} = \text{COMBINE}^{(k)}\left(h_v^{(k-1)}, \text{AGGREGATE}^{(k)}\left(\left\{\left(h_v^{(k-1)}, h_u^{(k-1)}, e_{uv}\right) : u \in \mathcal{N}(v)\right\}\right)\right), \quad (1)$$

where $h_v^{(k)}$ is the feature vector of node $v$ at the $k$-th iteration/layer (or patch representation centered at node $i$), $e_{uv}$ is the feature vector of the edge between $u$ and $v$, and $\mathcal{N}(v)$ are neighborhoods to node $v$. $h_v^{(0)}$ is often initialized as node features. READOUT can be a simple permutation invariant function such as averaging or a more sophisticated graph-level pooling function Ying et al. (2018); Zhang et al. (2018).

We seek to obtain graph representations by maximizing the mutual information between graph-level and patch-level representations. By doing so, the graph representations can learn to encode aspects of the data that are shared across all substructures. Assume that we are given a set of training samples $\mathbf{G} := \{G_j \in \mathbb{G}\}_{j=1}^N$ with empirical probability distribution $\mathbb{P}$ on the input space. Let $\phi$ denote the set of parameters of a $K$-layer graph neural network. After the first $k$ layers of the graph neural network, the input graph will be encoded into a set of patch representations $\{h_i^{(k)}\}_{i=1}^N$. Next, we summarize feature vectors at all depths of the graph neural network into a single feature vector that captures patch information at different scales centered at every node. Inspired by Xu et al. (2018b), we use

concatenation. That is,

$$h_\phi^i = \text{CONCAT}(\{h_i^{(k)}\}_{k=1}^K) \tag{2}$$

$$H_\phi(G) = \text{READOUT}(\{h_\phi^i\}_{i=1}^N) \tag{3}$$

where $h_\phi^i$ is the summarized patch representation centered at node $i$ and $H_\phi(G)$ is the global representation after applying READOUT. Note that here we slightly abuse the notation of $h$.

We define our mutual information (MI) estimator on global/local pairs, maximizing the estimated MI over the given dataset $\mathbf{G} := \{G_j \in \mathbb{G}\}_{j=1}^N$:

$$\hat{\phi}, \hat{\psi} = \arg\max_{\phi,\psi} \sum_{G \in \mathbf{G}} \frac{1}{|G|} \sum_{u \in G} I_{\phi,\psi}(\vec{h}_\phi^u; H_\phi(G)). \tag{4}$$

$I_{\phi,\psi}$ is the mutual information estimator modeled by discriminator $T_\psi$ and parameterized by a neural network with parameters $\psi$. We use the Jensen-Shannon MI estimator (following the formulation of Nowozin et al. (2016)),

$$I_{\phi,\psi}(h_\phi^i(G); H_\phi(G)) :=$$
$$\mathbb{E}_{\mathbb{P}}[-\text{sp}(-T_{\phi,\psi}(\vec{h}_\phi^i(x), H_\phi(x)))] - \mathbb{E}_{\mathbb{P}\times\tilde{\mathbb{P}}}[\text{sp}(T_{\phi,\psi}(\vec{h}_\phi^i(x'), H_\phi(x)))] \tag{5}$$

where $x$ is an input sample, $x'$ (negative sample) is an input sampled from $\tilde{\mathbb{P}} = \mathbb{P}$, a distribution identical to the empirical probability distribution of the input space, and $\text{sp}(z) = \log(1 + e^z)$ is the softplus function. In practice, we generate negative samples using all possible combinations of global and local patch representations across all graph instances in a batch.

Since $H_\phi(G)$ is encouraged to have high MI with patches that contain information at all scales, this favours encoding aspects of the data that are shared across patches and aspects that are shared across scales. The algorithm is illustrated in Fig. 1.

It should be noted that our model is similar to Deep Graph Infomax (DGI) Veličković et al. (2018), a model proposed for learning unsupervised node embeddings. However, there are important design differences due to the different problems that we are focusing on. First, in DGI they use random sampling to obtain negative samples due to the fact that they are mainly focusing on learning node embeddings on a graph. However, contrastive methods require a large number of negative samples to be competitive Hjelm et al. (2018), thus the use of batch-wise generation of negative samples is crucial as we are trying to learn graph embeddings given many graph instances. Second, the choice of graph convolution encoders is also crucial. We use GIN Xu et al. (2018a) while DGI uses GCN Kipf & Welling (2016) as GIN provides a better inductive bias for graph level applications. Graph neural network designs should be considered carefully so that graph representations can be discriminative towards other graph instances. For example, we use sum over mean for READOUT and that can provide important information regarding the size of the graph.

## 3.3 SEMI-SUPERVISED INFOGRAPH

Based on the previous unsupervised model, a straightforward way to do semi-supervised property prediction on graphs is to combine the purely supervised loss and the unsupervised objective function which acts as a regularization term. In doing so, the model is trained to predict properties for the labeled dataset while keeping a rich discriminative intermediate representation learned from both the labeled and the unlabeled dataset. That is, we try to minimize the following objective function:

$$L_{\text{total}} = \sum_{i=1}^{|\mathbb{G}^L|} L_{\text{supervised}}(y_\phi(G_i), o_i) + \lambda \sum_{j=1}^{|\mathbb{G}^L|+|\mathbb{G}^U|} L_{\text{unsupervised}}(h_\phi(G_j); H_\phi(G_j)) \tag{6}$$

where $L_{\text{supervised}}(y_\phi(G_i), o_i)$ is defined as the loss function of graph $G_i$ that measures the discrepancy between the classifier output $y_\phi(G_i)$ and the true output $o_i$. $L_{\text{unsupervised}}(h_\phi(G_j); H_\phi(G_j))$ is the

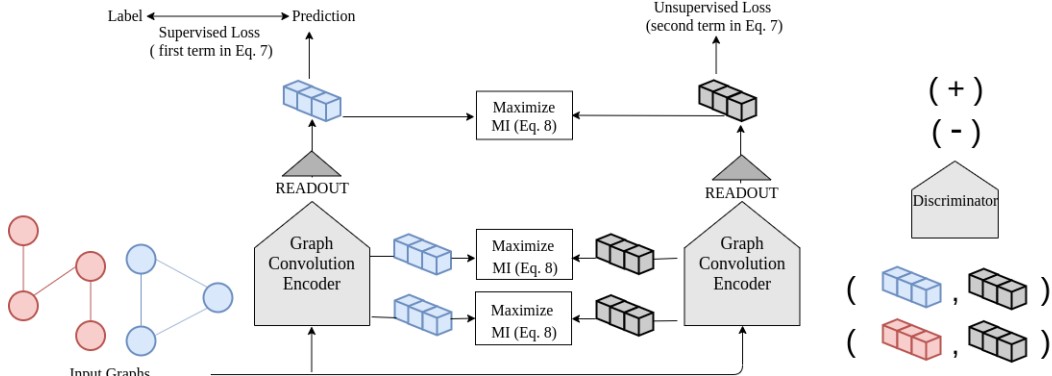

Figure 2: Illustration of the semi-supervised version of InfoGraph (InfoGraph*). There are two separate encoders with the same architecture, one for the supervised task and the other trained using both labeled and unlabeled data with an unsupervised objective (eq. equation 4). We encourage the mutual information of the two representations learned by the two encoders to be high by deploying a discriminator that takes a pair of representation as input and determines whether they are from the same input graph.

unsupervised InfoGraph loss term as defined in eq. equation 4 that can be optimized using both labeled and unlabeled data. The hyper-parameter $\lambda$ controls the relative weight between the purely supervised and the unsupervised loss. The intuition behind this is that the model will benefit from learning a good representation from the large amount of unlabeled data while learning to predict the corresponding supervised label.

However, supervised tasks and unsupervised tasks may favor different information or a different semantic space. Simply combining the two loss functions using the same encoder may lead to "negative transfer" [1] (Pan & Yang, 2009; Rosenstein et al., 2005). We propose a simple way to alleviate this problem: we deploy two encoder models: the encoder on the labelled data (supervised encoder) and the encoder on the unlabelled data (unsupervised encoder). For transferring the learned representations from the unsupervised encoder to the supervised encoder, we define a loss term that encourages the representations learned by the two encoders to have high mutual information, *at all levels of representations* (third term of Eq. 8). Formally, let $\varphi$ denote the set of parameters of another $K$-layered graph neural network, identical to the one parameterized by $\phi$, and let $\lambda$ be a tunable hyper-parameter, $H_\phi^k(G)$, $H_\varphi^k(G)$ be global encoder representations of the graph $G$ at encoder layer $k$, then total loss function can be defined as follows:

$$L_{\text{total}} = \sum_{i=1}^{|\mathbb{G}^L|} L_{\text{supervised}}(y_\phi(G_i), o_i) + \sum_{j=1}^{|\mathbb{G}^L|+|\mathbb{G}^U|} L_{\text{unsupervised}}(h_\varphi(G_j); H_\varphi(G_j)) \tag{7}$$

$$- \lambda \sum_{j=1}^{|\mathbb{G}^L|+|\mathbb{G}^U|} \frac{1}{|G_j|} \sum_{k=1}^{K} I(H_\phi^k(G_j); H_\varphi^k(G_j)). \tag{8}$$

Notice that this formulation can be seen as a special instance of the *student-teacher* framework. However, unlike the recent *student-teacher* methods for semi-supervised learning (Laine & Aila, 2016; Tarvainen & Valpola, 2017; Verma et al., 2019b), which enforce the predictions of the student model to be similar to the teacher model, we enforce the transfer of knowledge from the teacher model to the student model via mutual-information maximization at various levels of representations. In practice, to reduce the computation overhead introduced by the third term of Eq 8, instead of enforcing the mutual-information maximization over all the layers of the encoders, at each training

---

[1]We slightly abuse this term in this paper as it usually refers to transferring knowledge from a less related source and thus may hurt the target performance.

update, we enforce mutual-information maximization on a randomly chosen layer of the encoder (Verma et al., 2019a).

In our semi-supervised experiments, we refer to the naive method using the objective function given in eq. equation 6 as InfoGraph. We refer to the method that uses two separate encoders and employ the objective function given in eq. equation 8 as InfoGraph*. InfoGraph* is fully summarized in Figure 3.

## 4 EXPERIMENTS

We evaluate the effectiveness of the graph-level representation learned by InfoGraph on downstream graph classification tasks and on semi-supervised molecular property prediction tasks.

### 4.1 DATASETS

For graph classification, we conduct experiments on 6 well-known benchmark datasets: MUTAG, PTC, REDDIT-BINARY, REDDIT-MULTI-5K, IMDB-BINARY, and IMDB-MULTI (Yanardag & Vishwanathan (2015)). For semi-supervised learning tasks, we use the publicly available QM9 dataset Ramakrishnan et al. (2014). Additional details of the datasets can be found in Appendix B.

### 4.2 BASELINES

For graph classification, we used 6 state-of-the-art graph kernels for comparison: Random Walk (RW) Gärtner et al. (2003), Shortest Path Kernel (SP) Borgwardt & Kriegel (2005), Graphlet Kernel (GK) Shervashidze et al. (2009), Weisfeiler-Lehman Sub-tree Kernel (WL) Shervashidze et al. (2011), Deep Graph Kernels (DGK) Yanardag & Vishwanathan (2015), and Multi-Scale Laplacian Kernel (MLG) Kondor & Pan (2016). Aside from graph kernels, we also compare with 3 unsupervised graph-level representation learning methods: node2vec Grover & Leskovec (2016), sub2vec Adhikari et al. (2018), and graph2vec Narayanan et al. (2017). Node2vec is a neural embedding framework that learns feature representations of individual nodes in graphs and we aggregate node embeddings to obtain graph embeddings.

For semi-supervised tasks, aside from comparing the results with the fully supervised results, we also compare our results with a state-of-the-art semi-supervised method: Mean Teachers Tarvainen & Valpola (2017).

### 4.3 EXPERIMENT CONFIGURATION

For graph classification tasks, we adopt the same procedure of previous works Niepert et al. (2016); Verma & Zhang (2017); Yanardag & Vishwanathan (2015); Zhang et al. (2018) to make a fair comparison and used 10-fold cross validation accuracy to report the classification performance. Experiments are repeated 5 times. We report results from previous papers with the same experimental setup if available. If results are not previously reported, we implement them and conduct a hyper-parameter search according to the original paper. For node2vec Grover & Leskovec (2016), we took the result from Narayanan et al. (2017) but we did not run it on all datasets as the implementation details are not clear in the paper. For Deep Graph Kernels, we report the best result out of Deep WL Kernels, Deep GK Kernels, and Deep RW Kernels. For sub2vec, we report the best result out of its two variants: *sub2vec-N* and *sub2vec-S*. For all methods, the embedding dimension is set to 512 and parameters of downstream classifiers are independently tuned using cross validation on training folds of data. The best average classification accuracy is reported for each method. The classification accuracies are computed using LIBSVM Chang & Lin (2011), and the $C$ parameter was selected from $\{10^{-3}, 10^{-2}, \ldots, 10^2, 10^3\}$.

The QM9 dataset has 130462 molecules in it. We adopt similar experimental settings as traditional semi-supervised methods Tarvainen & Valpola (2017); Laine & Aila (2016); Miyato et al. (2018). We randomly chose 5000 samples as labeled samples for training and another 10000 as validation samples, 10000 samples for testing, and use the rest as unlabeled training samples. Note that we use the exact same split when running the supervised model and the semi-supervised model. We use the validation set to do model selection and we report scores on the test set. All targets were normalized

| Dataset | MUTAG | PTC-MR | RDT-B | RDT-M5K | IMDB-B | IMDB-M |
|---|---|---|---|---|---|---|
| (No. Graphs) | 188 | 344 | 2000 | 4999 | 1000 | 1500 |
| (No. classes) | 2 | 2 | 2 | 5 | 2 | 3 |
| (Avg. Graph Size) | 17.93 | 14.29 | 429.63 | 508.52 | 19.77 | 13.00 |
| Graph Kernels | | | | | | |
| RW | $83.72 \pm 1.50$ | $57.85 \pm 1.30$ | OMR | OMR | $50.68 \pm 0.26$ | $34.65 \pm 0.19$ |
| SP | $85.22 \pm 2.43$ | $58.24 \pm 2.44$ | $64.11 \pm 0.14$ | $39.55 \pm 0.22$ | $55.60 \pm 0.22$ | $37.99 \pm 0.30$ |
| GK | $81.66 \pm 2.11$ | $57.26 \pm 1.41$ | $77.34 \pm 0.18$ | $41.01 \pm 0.17$ | $65.87 \pm 0.98$ | $43.89 \pm 0.38$ |
| WL | $80.72 \pm 3.00$ | $57.97 \pm 0.49$ | $68.82 \pm 0.41$ | $46.06 \pm 0.21$ | $72.30 \pm 3.44$ | $46.95 \pm 0.46$ |
| DGK | $87.44 \pm 2.72$ | $60.08 \pm 2.55$ | $78.04 \pm 0.39$ | $41.27 \pm 0.18$ | $66.96 \pm 0.56$ | $44.55 \pm 0.52$ |
| MLG | $87.94 \pm 1.61$ | $\mathbf{63.26 \pm 1.48}$ | > 1 Day | > 1 Day | $66.55 \pm 0.25$ | $41.17 \pm 0.03$ |
| Other Unsupervised Methods | | | | | | |
| node2vec | $72.63 \pm 10.20$ | $58.58 \pm 8.00$ | - | - | - | - |
| sub2vec | $61.05 \pm 15.80$ | $59.99 \pm 6.38$ | $71.48 \pm 0.41$ | $36.68 \pm 0.42$ | $55.26 \pm 1.54$ | $36.67 \pm 0.83$ |
| graph2vec | $83.15 \pm 9.25$ | $60.17 \pm 6.86$ | $75.78 \pm 1.03$ | $47.86 \pm 0.26$ | $71.1 \pm 0.54$ | $\mathbf{50.44 \pm 0.87}$ |
| **InfoGraph** | $\mathbf{89.01 \pm 1.13}$ | $61.65 \pm 1.43$ | $\mathbf{82.50 \pm 1.42}$ | $\mathbf{53.46 \pm 1.03}$ | $\mathbf{73.03 \pm 0.87}$ | $49.69 \pm 0.53$ |

Table 1: Classification accuracy on 6 datasets. The result in **bold** indicates the best reported classification accuracy. The top half of the table compares results with various graph kernel approaches while bottom half compares results with other state-of-the-art unsupervised graph representation learning methods. '> 1 day' represents that the computation exceeds 24 hours. 'OMR' is out of memory error.

| Target | Mu (0) | Alpha (1) | HOMO (2) | LUMO (3) | Gap (4) | R2 (5) | ZPVE(6) | U0 (7) | U (8) | H (9) | G(10) | Cv (11) |
|---|---|---|---|---|---|---|---|---|---|---|---|---|
| MAE | 0.3201 | 0.5792 | 0.0060 | 0.0062 | 0.0091 | 10.0469 | 0.0007 | 0.3204 | 0.2934 | 0.2722 | 0.2948 | 0.2368 |

| Semi-Supervised | Error Ratio | | | | | | | | | | | |
|---|---|---|---|---|---|---|---|---|---|---|---|---|
| Mean-Teachers | 1.09 | 1.00 | **0.99** | 1.00 | **0.97** | 0.52 | 0.77 | 1.16 | 0.93 | 0.79 | 0.86 | 0.86 |
| InfoGraph | 1.02 | 0.97 | 1.02 | **0.99** | 1.01 | 0.71 | 0.96 | 0.85 | 0.93 | 0.93 | 0.99 | 1.00 |
| InfoGraph* | **0.99** | **0.94** | **0.99** | **0.99** | 0.98 | **0.49** | **0.52** | **0.44** | **0.58** | **0.57** | **0.54** | **0.83** |

Table 2: Results of semi-supervised experiments on QM9 dataset. The result in **bold** indicates the best performance. The top half of the table shows the mean absolute error (MAE) of the supervised model. The bottom half shows the error ratio (with respect to supervised result) of the semi-supervised models using the same underlying model. Lower scores are better and values less than 1.0 indicate better performance than the supervised baseline.

to have mean 0 and variance 1. We minimize the mean squared error between the model output and the target, although we evaluate mean absolute error.

## 4.4 MODEL CONFIGURATION

For the unsupervised experiments, we use the Graph Isomorphism Network (GIN) Xu et al. (2018a). For the semi-supervised experiments, we adopt the same model as in Gilmer et al. (2017) (enn-s2s). As recommended in Oliver et al. (2018), we use the exact same underlying model architecture when comparing semi-supervised learning approaches as our goal is not to produce state-of-the-art results, but instead to provide a rigorous comparative analysis in a common framework. In both scenarios, models were trained using SGD with the Adam optimizer. We use Pytorch Paszke et al. (2017) and the Pytorch Geometric Fey & Lenssen (2019) libraries for all our experiments. For detailed hyper-parameter settings and architecture detail of the discriminator, see Appendix C.

## 5 RESULTS

The results of evaluating unsupervised graph level representations using downstream graph classification tasks are presented in Table 1. We show results from six methods including three state-of-the-art graph kernel methods: WL Shervashidze et al. (2011), DGK Yanardag & Vishwanathan (2015), and MLG Kondor & Pan (2016). While these kernel methods perform well on individual datasets, none of them are competitive across all of the datasets. Additionally, MLG suffers from a long run time and take more than 24 hours to run on the two larger benchmark datasets. We find that InfoGraph

outperforms all of these baselines on 4 out of 6 of the datasets. In the other 2 datasets, InfoGraph still has very competitive performance.

The results of the semi-supervised learning experiments on the molecular property prediction task are presented in Table 2. We observe that by simply combining the supervised objective with the unsupervised infomax objective (InfoGraph) obtains better performance compared to the purely supervised models on 7 out of 12 of the targets. However, in 1 out of 12 targets it does not obtain better performance and in 4 out of 12 targets, it results in poorer performance. This "negative transfer" effect may be caused by the fact that the supervised objective and the unsupervised objective favor different information or different latent semantic space. This effect is alleviated with InfoGraph*, our modified version of InfoGraph for semi-supervised learning. InfoGraph* improves over the supervised model in all the 12 targets. InfoGraph* obtains the best result on 11 targets while the Mean Teacher method obtains the best results on 2 targets (with one overlap). However, the Mean Teacher model yields worse performance on 2 targets when compared to the supervised result.

## 6    CONCLUSION AND FUTURE WORK

In this paper, we propose InfoGraph to learn unsupervised graph-level representations and InfoGraph* for semi-supervised learning. We conduct experiments on graph classification and molecular property prediction tasks to evaluate these two methods. Experimental results show that InfoGraph and InfoGraph* are both very competitive with state-of-the-art methods. There are many research works on semi-supervised learning on image data, but few of them focus on semi-supervised learning for graph structured data. In the future, we aim to explore semi-supervised frameworks designed specifically for graphs.

### ACKNOWLEDGMENTS

We would like to thank Shengchao Liu and Weihua Hu for the extremely helpful discussions and comments.

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

## A  Related Work

### A.1  Graph Kernels

Popular graph kernels are graphlets Pržulj (2007); Shervashidze et al. (2009), random walk and shortest path kernels Kashima et al. (2003); Borgwardt & Kriegel (2005), and the Weisfeiler-Lehman subtree kernel Shervashidze et al. (2011). Furthermore, deep graph kernels Yanardag & Vishwanathan (2015), graph invariant kernels Orsini et al. (2015), optimal assignment graph kernels Kriege et al. (2016) and multiscale Laplacian graph kernels Kondor & Pan (2016) have been proposed with the goal to redefine kernel functions to appropriately capture sub-structural similarity at different levels. Another line of research in this area focuses on efficiently computing these kernels either through exploiting certain structural dependencies, or via approximations/randomization Feragen et al. (2013); de Vries (2013); Neumann et al. (2012).

### A.2  Contrastive Methods

Contrastive methods are central many popular word-embedding methods Collobert & Weston (2008); Mnih & Kavukcuoglu (2013); Mikolov et al. (2013b). Word2vec Mikolov et al. (2013a) is an unsupervised algorithm which obtains word representations by using the representations to predict context words (the words that surround it). Doc2vec Le & Mikolov (2014) is an extension of the continuous Skip-gram model that predicts representations of words from that of a document containing them. Researchers extended many of these unsupervised language models to learn representations of graph-structured input Adhikari et al. (2018); Narayanan et al. (2017). For example, *graph2vec* Narayanan et al. (2017) extends Doc2vec to arbitrary graphs. Intuitively, for graph2vec a graph and the rooted subgraphs in it correspond to a document and words in a paragraph vector, respectively. One of the technical contributions of the paper is using the Weisfeiler-Lehman relabelling algorithm Weisfeiler & Lehman (1968); Shervashidze et al. (2011) to enumerate all rooted subgraphs up to a specified depth. AWE (Anonymous Walk Embeddings) Ivanov & Burnaev (2018) is another method based on CBOW framework. instead of using rooted subgraphs as words like graph2vec, AWE considers anonymous walk embeddings for the same source node as co-occurring words. InfoGraph has the two advantages when compared with these methods. First, InfoGraph learns representations directly from data instead of utilizing hand-crafted procedures (i.e. Weisfeiler-Lehman relabelling algorithm in graph2vec and random walkw in AWE). Second, InfoGraph has a clear objective that can be easily combined with other objectives. For example, InfoGraph*, the semi-supervised method that we proposed.

## B  Semi-Supervised Learning

Here we discuss the most common class of SSL methods which involve adding an additional loss term to the training of a neural network as they are pragmatic and are currently the state-of-the-art on image classification datasets.

**Entropy Minimization (EntMin):** EntMin Grandvalet & Bengio (2005) adds a loss term applied that encourages the network to make "confident" (low-entropy) predictions for all unlabeled examples, regardless of their class.

**Pseudo-Labeling:** Pseudo-labeling Lee (2013) proceeds by producing "pseudo-labels" for unlabeled input data points using the prediction function itself over the course of training. Pseudo-labels which have a corresponding class probability that is larger than a predefined threshold are used as targets for a the standard supervised loss function.

**Π-Model:** Neural networks can produce different outputs for the same input while common regularization techniques such as data augmentation, dropout, and adding noise are applied. Π-Model Laine & Aila (2016); Sajjadi et al. (2016) adds a loss term which encourages the distance between a network's output for different passes of unlabeled data through the network to be small.

**Virtual Adversarial Training:** Instead of relying on the built-in stochasticity as in Π-Model, Virtual Adversarial Training (VAT) Miyato et al. (2018) directly approximates a tiny perturbation to add to the input which would most significantly affect the output of the prediction function. This pertubation can be approximated with an extra back-propagation for each optimization step.

**Mean Teacher:** A difficulty with the $\Pi$-model approach is that it relies on a potentially unstable "target" prediction, namely the second stochastic network prediction which can rapidly change over the course of training. As a result, Tarvainen & Valpola (2017) proposed to obtain a more stable target output for unlabeled data by setting the target to predictions made using an exponential moving average of parameters from previous training steps.

## C  DATASETS

### C.1  GRAPH CLASSIFICATION DATASETS

MUTAG contains 188 mutagenic aromatic and heteroaromatic nitro compounds with 7 different discrete labels. PTC is a dataset of 344 different chemical compounds that have been tested for carcinogenicity in male and female rats. This dataset has 19 discrete labels. IMDB-BINARY and IMDB-MULTI are movie collaboration datasets. Each graph corresponds to an ego-network for each actor/actress, where nodes correspond to actors/actresses and an edge is drawn between two actors/actresses if they appear in the same movie. Each graph is derived from a pre-specified genre of movies, and the task is to classify the genre graph it is derived from. REDDIT-BINARY and REDDIT-MULTI5K are balanced datasets where each graph corresponds to an online discussion thread and nodes correspond to users. An edge was drawn between two nodes if at least one of them responded to another's comment. The task is to classify each graph to the community or subreddit that it belongs to.

### C.2  QM9

All molecules in the dataset consist of Hydrogen (H), Carbon (C), Oxygen (O), Nitrogen (N), and Flourine (F) atoms and contain up to 9 non-Hydrogen atoms. In all, this results in about 134,000 drug-like organic molecules that span a wide range of chemical compositions and properties. A total of 12 interesting and fundamental chemical properties are pre-computed for each molecule. For a detailed description of the properties in the QM9 dataset, see section 10.2 of Gilmer et al. (2017).

## D  MODEL CONFIGURATION

For the unsupervised experiments, we use the Graph Isomorphism Network (GIN) Xu et al. (2018a). GNN layers are chosen from $\{4, 8, 12\}$. Initial learning rate is chosen from the set $\{10^{-2}, 10^{-3}, 10^{-4}\}$. The number of epochs are chosen from $\{10, 20, 100\}$. The batch size is set to 128.

For the semi-supervised experiments, the number of set2set computations is set to 3. Model were trained with an initial learning rate 0.001 for 500 epochs with a batch size 20. For the supervised case, the weight decay is chosen from $\{0, 10^{-3}, 10^{-4}\}$. For InfoGraph and InfoGraph*, $\lambda$ is chosen from $\{10^{-3}, 10^{-4}, 10^{-5}\}$.

The discriminator scores global-patch representation pairs by passing two representations to different non-linear transformations and then takes the dot product of the two transformed representations. Both non-linear transformations are parameterized by 3-layered feed-forward neural networks with jumping connections. Following each linear layer is a ReLU activation function.

# E    CONVERGENCE PLOT

To prove that the objective Eq.8 with multiple loss terms can be optimized, we provide a convergence plot of InfoGraph*. We can see that the three loss terms all converge after around 150 epochs of training.

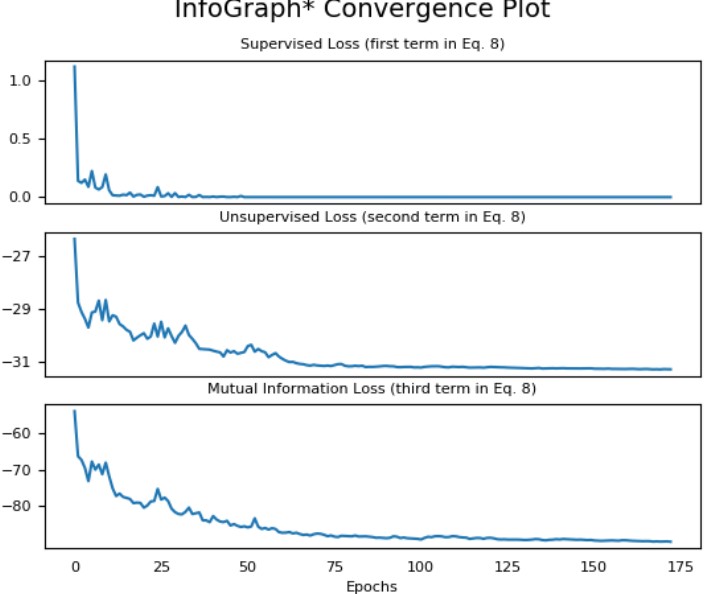

Figure 3: Convergence plot of InfoGraph* on QM9 target 7.

