# OpenReview forum: "InfoGraph: Unsupervised and Semi-supervised Graph-Level Representation Learning via Mutual Information Maximization"
_ICLR.cc/2020/Conference — Accept (Spotlight)_

### Official Review · AnonReviewer2 · 2019-10-28
**Official Blind Review #2**

**Rating:** 6

**Review:**

In this paper, the authors propose a graph-level representation, which extends the existing node-level representation learning mechanism. Besides, both unsupervised and semi-supervised learning are leveraged for InfoGraph and InfoGraph*, receptively. The authors naturally apply Deep Graph Infomax, a contrastive representation learning method, for the whole graph level instead of the previous node embedding learning. The experiments on graph classification and molecular property prediction indicate the effectiveness, even compared to the supervised methods.

The learned graph-level representation looks good to me. Instead of some heuristics based graph-level pooling, the proposed method automatically figure out the best way to produce the fixed-length feature vector in a data-driven approach. Such motivation is pretty reasonable and natural for me. Also, the usage of both unsupervised and semi-supervised learning procedure is well-motivated. Overall speaking, the paper is well written. The definitions of problems, the details of methods, and the settings of experiments are clear to me.

I have some questions and suggestions for the authors:
1. The overall writing looks good to me. Besides, It could be much better if the authors could provide more and better illustrations for the method. Both figures 1 and 2 are not that informative to me, to be honest. I know it could hard to visualize the graph-level representation, but it worth it. There are many steps and equations in the paper, and the illustration could play an essential role in putting all the steps together to demonstrate the big picture.

2. The authors carefully discuss the difference between this submission and concurrent work (Information Maximizing Graph Network) or existing work (Deep Graph Infomax), which helps a lot for the reader to understand the literature better. It could be much better if this submission could be more self-contained. For example, for the semi-supervised learning setting, the authors let the reader read some external papers. I am suggesting that a small section in the appendix could make life much more comfortable.

**Experience Assessment:**

I have read many papers in this area.

**Review Assessment: Checking Correctness Of Derivations And Theory:**

I assessed the sensibility of the derivations and theory.

**Review Assessment: Checking Correctness Of Experiments:**

I assessed the sensibility of the experiments.

**Review Assessment: Thoroughness In Paper Reading:**

I read the paper at least twice and used my best judgement in assessing the paper.

---

> ### Author Response · Authors · 2019-11-10
> **Response**
>
> Thank you for the review and feedback!
>
> 1. We updated the figures as you suggested. we hope the updated version can better demonstrate the big picture. In the figure, we provide connections between components in the framework and equations in the paper.
>
> 2. We added a section of a quick overview of semi-supervised learning (SSL) to the appendix. We hope that this will make our submission more self-contained.

---

### Official Review · AnonReviewer1 · 2019-10-28
**Official Blind Review #1**

**Rating:** 6

**Review:**

The paper presents an unsupervised method for graph embedding. The authors seek to obtain graph representations by maximizing the mutual information between graph-level and patch-level
representations. They also consider a semi-supervised task when the Mutual Information-based criterion has an additional term which quantifies a classification error, obtained when constructing a classifier based on the obtained graph representations.

Despite having good experimental results, the proposed approach is rather a mix of previous works and hence not novel.

In particular, the main building block of the embedding algorithm, the target functional based on mutual information, was borrowed from Deep Graph Informax paper. The differences, listed by the authors, are only of technical nature. Advantage of using it for unlabeled data is poorly motivated: why we can learn smth useful when maximizing the mutual information between graph-level and patch-level representations obtained via GNN? What if patch-level representations are not sufficiently characteristic to have anything in common with the graph?

There is no discussion of [1], which uses CBOW framework, has theoretical properties, and produces good results in experiments. There is no comparison with GNN models such as [2].

Minor comments: please, correct fonts - they are different in formulas 6,7 and 5

I would be more interested to see explanation of the obtained results for each particular dataset (e.g. why MUTAG has 89% accuracy and PTC 61%); what so different about dataset and whether we reached a limit on most commonly used datasets.

[1] Anonymous Walk Embeddings? ICML 2018, Ivanov et. al.
[2] How Powerful are Graph Neural Networks? ICLR 2019, Xu et. al.

**Experience Assessment:**

I have published one or two papers in this area.

**Review Assessment: Checking Correctness Of Derivations And Theory:**

N/A

**Review Assessment: Checking Correctness Of Experiments:**

I assessed the sensibility of the experiments.

**Review Assessment: Thoroughness In Paper Reading:**

I read the paper at least twice and used my best judgement in assessing the paper.

---

> ### Author Response · Authors · 2019-11-10
> **Response**
>
> Thank you for the review and feedback! For minor errors, we have fixed them.
> In the following, we address the concerns point by point:
>
> 1. InfoGraph and Deep Graph Infomax are indeed similar in the way of exploiting mutual information maximization for learning data representation. However, Deep Graph Infomax focused on learning node representations whereas InfoGraph are the first model to learn the graph-level representation by using mutual information maximization. Furthermore, we proposed InfoGraph*, which is a novel semi-supervised graph learning framework that outperforms SOTA MeanTeachers baseline model.
>
> 2. Q: Why we can learn smth useful when maximizing the mutual information between graph-level and patch-level representations obtained via GNN?  What if patch-level representations are not sufficiently characteristic to have anything in common with the graph?
>
> Here, we provide more motivation for maximizing the mutual information between the graph-level and patch-level representations. Patch-level representations are obtained through a trainable GCN encoder and the Graph-level representation is the output of the READOUT function whose input is a set of patch-level representations. Since the global representation is a feature vector of limited capacity/dimension, it is forced to be selective to what type of information is passed through the encoder and preserved at the graph-level vector. Indeed, some patch-level representations may not have sufficiently characteristic to have anything in common with the graph-level representation. However, InfoGraph will force the encoder to learn patch-level representations in a way where many patch-level representations sufficiently common/related with the graph-level representation so that the mutual information between the graph-level representation and all patch-level representations is maximized. In other words, the encoder is forced to favor encoding aspects of the data that are shared across patches.
>
> 3. We have discussed unsupervised graph-level representation learning methods based on CBOW framework in Appendix A. [1] can be viewed as a revision of graph2vec [4] where instead of using rooted subgraphs as words, anonymous random walks for the same source node are considered as co-occurring words. InfoGraph has the following advantages when compared with these methods.
> a. InfoGraph learns representations directly from data instead of utilizing hand-crafted procedures (i.e. Weisfeiler-Lehman relabelling algorithm in graph2vec and anonymous random walk in [1])
> b. InfoGraph has a clear objective that can be easily combined with other objective. For example, InfoGraph*, the semi-supervised method that we proposed.
> We have added this discussion to appendix A.
>
> 4. [2] proposed a variant of the GNN called Graph Isomorphism Networks (GINs). They evaluated their model in a supervised setting. However, as suggested by the title of our paper, we aim to learn graph-level representations in unsupervised (InfoGraph) and semi-supervised (InfoGraph*) fashion. Thus, InfoGraph/InfoGraph* are not directly comparable to [3]. In fact, we adopt GINs in our framework, as mentioned in section 4.4.
>
> 5. Aside from the number of classes, which obviously affects the resulting accuracy significantly, there are many other factors that vary a lot between datasets that can cause discrepancies in terms of accuracies. For example, MUTAG is a dataset of chemical compounds, where identifying scaffolds and meaningful substructures are the key to making accurate predictions, while for IMDB, a dataset of ego networks, the size of the graph and the degree distribution may be more important. This is also the reason why there are graph kernels tailored to certain domains of data. For a more detailed description and analysis, refer to [3]. Also, these datasets and their source/explanation can be found here: https://ls11-www.cs.tu-dortmund.de/staff/morris/graphkerneldatasets.
>
>
> [1] Anonymous Walk Embeddings? ICML 2018, Ivanov et. al.
> [2] How Powerful are Graph Neural Networks? ICLR 2019, Xu et. al.
> [3] Effective graph classification based on topological and label attributes. Statistical Analysis and Data Mining: The ASA Data Science Journal, 5(4), pp.265-283. Li, G., Semerci, M., Yener, B. and Zaki, M.J., 2012.
> [4] graph2vec: Learning distributed representations of graphs. Narayanan, A., Chandramohan, M., Venkatesan, R., Chen, L., Liu, Y. and Jaiswal, S., 2017.

---

> > ### Comment · AnonReviewer1 · 2019-11-13
> > **Thanks for addressing comments**
> >
> > I am OK with authors' comments.
> > I think that the paper deserves to be published.
> > The authors improved the presentation and addressed my comments.
> > Therefore, I can increase the grade.

---

### Official Review · AnonReviewer3 · 2019-10-31
**Official Blind Review #3**

**Rating:** 6

**Review:**

The paper presents a new graph representation learning method for the whole graph under both unsupervised and semi-supervised setting. Different from existing ones using graph kernel, or graph2vec, the proposed InfoGraph is able to extract graph-level representation with fixed-length features that are generalized well. Basically, InfoGraph is parameterized by graph neural networks, but guided by mutual information loss. Experiments on both unsupervised and semi-supervised experiments on popular benchmarks demonstrate the effectiveness of InfoGraph and InfoGraph*

*  The paper is well written and easy to follow, and the research problem is of great value in different fields.

* In general, the novelty of this paper is ok, but it’s partially based on Deep InfoMax (DIM) published recently. This may undermine the novelty of this paper somehow.

* Authors change the fonts in equation from italic to non-italic in Eq. (6), please make sure to use one format throughout the paper.

* Why Jensen-Shannon MI estimator is used in Eq. (5) instead of other estimators for MI, and any more explanations or motivations here?

* Eq. (7) and (8) in facts jointly optimize between two networks \phi and \varphi, but little details about the optimization have been exposed. Also, we are not sure if the loss from the two models can converge finally. Better to show some qualitative results and analysis.

* READOUT seems to play a critical role in building the global representation, however, it is unclear if other READOUT function will work well, and why the current one is feasible. Please explain.

* In semi-supervised setting, it seems InfoGraph* is comparable to the SOTA MeanTeachers model on 12 targets. I am not sure if this can reflect the true performance of the proposed model. Maybe another dataset will be able to highlight the superiority.


**Experience Assessment:**

I have published one or two papers in this area.

**Review Assessment: Checking Correctness Of Derivations And Theory:**

I assessed the sensibility of the derivations and theory.

**Review Assessment: Checking Correctness Of Experiments:**

I carefully checked the experiments.

**Review Assessment: Thoroughness In Paper Reading:**

I read the paper thoroughly.

---

> ### Author Response · Authors · 2019-11-10
> **Response**
>
> Thank you for the review and feedback! For minor errors such as inconsistent fonts, we have fixed them.
> In the following, we address the concerns point by point:
>
> 1. We use Jensen-Shannon MI estimator mainly motivated by [1]. In [1], they mentioned that various MI estimators work but Jensen-Shannon MI estimator is more stable and provide better results. In their appendix, they showed the relationship between the Jensen-Shannon divergence (JSD) between the joint and the product of marginals and the pointwise mutual information.
>
> 2. To prove that the loss from two models can converge, we provided a convergence plot in the appendix.
>
> 3. In short, as recommended in [2], we we use sum instead of mean or max pooling because sum has more expressive/discriminate power over a multiset.
>
> 4. The paper is updated with new results obtained after the submission deadline. We improved our model by encouraging the representations learned by the two encoders to have high mutual information at all levels of representations (refer to the third term of Eq. 8). Currently, we outperform SOTA MeanTeachers model on 10 out of 12 targets.
>
> [1] Hjelm, R.D., Fedorov, A., Lavoie-Marchildon, S., Grewal, K., Bachman, P., Trischler, A. and Bengio, Y., 2018. Learning deep representations by mutual information estimation and maximization.
> [2] Xu, K., Hu, W., Leskovec, J. and Jegelka, S., 2018. How powerful are graph neural networks?

---

### Author Response · Authors · 2019-11-07
**Submission Revision 1: Summary of Changes**

At the beginning of the rebuttal period, the submission was updated with new results obtained after the submission deadline. Aside from correcting fonts, two changes were made:

(1) We updated the figures to better demonstrate the big picture. In the figure, we also provide connections between components in the framework and equations in the paper.

(2) A substantial change was made:
We improved InfoGraph* by encouraging the representations learned by the two encoders to have high mutual information at all levels of representations instead of one chosen level (refer to the third term of Eq. 8). In practice, to reduce the computation overhead introduced, we enforce mutual information maximization on a randomly chosen layer of the encoder at each training update (motivated by [1]). The results in Table 2 are updated. Currently, InfoGraph* improves over the supervised model in all the 12 targets.  InfoGraph* obtains the best result on 11 targets while the Mean Teacher method obtains the best results on 2 targets (with one overlap).

[1] Verma, V., Lamb, A., Beckham, C., Najafi, A., Mitliagkas, I., Courville, A., Lopez-Paz, D. and Bengio, Y., 2018. Manifold mixup: Better representations by interpolating hidden states.

---

### Decision · Program_Chairs · 2019-12-19

**Decision:**

Accept (Spotlight)

**Comment:**

This paper proposes a graph embedding method for the whole graph under both unsupervised and semi-supervised setting. It can extract a fixed length graph-level representation with good generalization capability. All reviewers provided unanimous rating of weak accept. The reviewers praise the paper is well written and is value to different fields dealing with graph learning. There are some discussions on the novelty of the approach, which was better clarified after the response from the authors. Overall this paper presents a new effort in the active topic of graph representation learning with potential large impact to multiple fields. Therefore, the ACs recommend it to be an oral paper.